# Large Margin Discriminant Dimensionality Reduction in Prediction Space

**Mohammad Saberian**
Netflix
esaberian@netflix.com

**Jose Costa Pereira**
INESCTEC
jose.c.pereira@inesctec.pt

**Can Xu**
Google
canxu@google.com

**Jian Yang**
Yahoo Research
jianyang@yahoo-inc.com

**Nuno Vasconcelos**
UC San Diego
nvasconcelos@ucsd.edu

## Abstract

In this paper we establish a duality between boosting and SVM, and use this to derive a novel discriminant dimensionality reduction algorithm. In particular, using the multiclass formulation of boosting and SVM we note that both use a combination of mapping and linear classification to maximize the multiclass margin. In SVM this is implemented using a pre-defined mapping (induced by the kernel) and optimizing the linear classifiers. In boosting the linear classifiers are pre-defined and the mapping (predictor) is learned through a combination of weak learners. We argue that the intermediate mapping, i.e. boosting predictor, is preserving the discriminant aspects of the data and that by controlling the dimension of this mapping it is possible to obtain discriminant low dimensional representations for the data. We use the aforementioned duality and propose a new method, Large Margin Discriminant Dimensionality Reduction (LADDER) that jointly learns the mapping and the linear classifiers in an efficient manner. This leads to a data-driven mapping which can embed data into any number of dimensions. Experimental results show that this embedding can significantly improve performance on tasks such as hashing and image/scene classification.

## 1 Introduction

Boosting and support vector machines (SVM) are widely popular techniques for learning classifiers. While both methods are maximizing the margin, there are a number of differences that distinguish them; e.g. while SVM selects a number of examples to assemble the decision boundary, boosting achieves this by combining a set of weak learners. In this work we propose a new duality between boosting and SVM which follows from their multiclass formulations. It shows that both methods seek a linear decision rule by maximizing the margin after transforming input data to an intermediate space. In particular, kernel-SVM (K-SVM) [39] first selects a transformation (induced by the kernel) that maps data points into an intermediate space, and then learns a set of linear decision boundaries that maximize the margin. This is depicted in Figure 1-bottom. In contrast, multiclass boosting (MCBoost) [34] relies on a set of pre-defined codewords in an intermediate space, and then learns a mapping to this space such that it maximizes the margin with respect to the boundaries defined by those codewords. See Figure 1-top. Therefore, both boosting and SVM follow a two-step procedure: (i) mapping data to some intermediate space, and (ii) determine the boundaries that separate the classes. There are, however, two notable differences: 1) while K-SVM aims to learn only the boundaries, MCBoost effort is on learning the mapping and 2) in K-SVM the intermediate space typically has infinite dimensions, while in MCBoost the space has $M$ or $M-1$ dimensions, where $M$ is the number of classes.

The intermediate space (called *prediction space*) in the exposed duality has some interesting properties. In particular, the final classifier decision is based on the representation of data points in this prediction space where the decision boundaries are linear. An accurate classification by these simple boundaries suggests that the input data points must be very-well separated in this space. Given that in the case of boosting this space has limited dimensions, e.g. $M$ or $M - 1$, this suggests that we can potentially use the predictor of MCBoost as a discriminant dimensionality reduction operator. However, the dimension of MCBoost is either $M$ or $M - 1$ which restricts application of this operator as a general dimensionality reduction operator. In

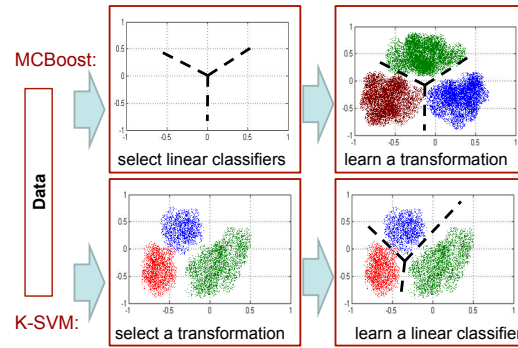

Figure 1: Duality between multiclass boosting and SVM.

addition, according to the proposed duality, each of K-SVM or Boosting optimizes only one of the two components, i.e. mapping and decision boundaries. Because of this, extra care needs to be put in manually choosing the *right* kernel in K-SVM; and in MCBoost, we may not even be able to learn a *good* mapping if we preset some bad boundaries.

We can potentially overcome these limitations by combining boosting and SVM to jointly learn both the mapping and linear classifiers for a prediction space of arbitrary dimension $d$. We note that this is not a straightforward merge of the two methods as this can lead to a computationally prohibitive method; e.g. imagine having to solve the quadratic optimization of K-SVM before each iteration of boosting. In this paper, we propose a new algorithm, *Large-mArgin Discriminant DimEnsionality Reduction* (LADDER), to efficiently implement this hybrid approach using a boosting-like method. LADDER is able to learn both the mapping and the decision boundaries in a margin maximizing objective function that is adjustable to any number of dimensions. Experiments show that the resulting embedding can significantly improve tasks such as hashing and image/scene classification.

**Related works:** This paper touches several topics such as dimensionality reduction, classification, embedding and representation learning. Due to space constraints we present only a brief overview and comparison to previous work.

Dimensionality reduction has been studied extensively. Unsupervised techniques, such as principal component analysis (PCA), non-negative matrix factorization (NMF), clustering, or deep auto-encoders, are conceptually simple and easy to implement, but may eliminate discriminant dimensions of the data and result in sub-optimal representations for classification. Discriminant methods, such as sequential feature selection techniques [31], neighborhood components analysis [11], large margin nearest neighbors [42] or maximally collapsing metric learning [37] can require extensive computation and/or fail to guarantee large margin discriminant data representations.

The idea of jointly optimizing the classifiers and the embedding has been extensively explored in embedding and classification literature, e.g. [7, 41, 45, 43]. These methods, however, typically rely on linear data transformation/classifier, requires more complex semi-definite programming [41] or rely on Error Correcting Output Codes (ECOC) approach [7, 45, 10] which has shown inferior performance compared to direct multiclass boosting methods [34, 27]. In comparison, we note that the proposed method (1) is able to learn a very non-linear transformation through boosting predictor, e.g. boosting deep decision trees; and, (2) relies on direct multiclass boosting that optimizes a margin enforcing loss function. Another example of jointly learning the classifiers and the embedding is multiple kernel learning (MKL) literature, e.g. [12, 36]. In these methods, a new kernel is learned as a linear combination of fixed basis functions. Compared with LADDER, 1) the basis functions are data-driven and not fixed, and 2) our method is also able to combine weak learners and form novel basis functions tailored for the current task. Finally, it is also possible to jointly learn the classifiers and embedding using deep neural networks. This, however, requires large number of training data and can be computationally very intensive. In addition the proposed LADDER method is a meta algorithm that can be used to further improve the deep networks, e.g. by boosting of the deep CNNs.

## 2 Duality of boosting and SVM

Consider an $M$-class classification problem, with training set $\mathcal{D} = \{(x_i, z_i)\}_{i=1}^n$, where $z_i \in \{1 \ldots M\}$ is the class of example $x_i$. The goal is to learn a real-valued (multidimensional) function $f(x)$ to predict the class label $z$ of each example $x$. This is formulated as the predictor $f(x)$ that minimizes the risk defined in terms of the expected loss $L(z, f(x))$:

$$R[f] = E_{X,Z}\{L(z, f(x))\} \approx \frac{1}{n} \sum_i L(z_i, f(x_i)). \tag{1}$$

Different algorithms vary in their choice of loss functions and numerical optimization procedures. The learned predictor has large margin if the loss $L(z, f(x))$ encourages large values of the classification margin. For binary classification, $f(x) \in \mathbb{R}$, $z \in \{1, 2\}$, the margin is defined as $\mathcal{M}(x_i, z_i) = y_i f(x_i)$, where $y_i = y(z_i) \in \{-1, 1\}$ is the *codeword* of class $z_i$. The classifier is then $F(x) = \mathcal{H}(sign[f(x)])$ where $\mathcal{H}(+1) = 1$ and $\mathcal{H}(-1) = 2$.

The extension to M-ary classification requires $M$ codewords. These are defined in a multidimensional space, i.e. as $y^k \in \mathbb{R}^d, k = 1 \ldots M$ where commonly $d = M$ or $d = M - 1$. The predictor is then $f(x) = [f_1(x), f_2(x) \ldots f_d(x)] \in \mathbb{R}^d$, and the margin is defined as

$$\mathcal{M}(x_i, z_i) = \frac{1}{2} \left[ \langle f(x_i), y^{z_i} \rangle - \max_{l \neq z_i} \langle f(x_i), y^l \rangle \right], \tag{2}$$

where $\langle \cdot, \cdot \rangle$ is the Euclidean dot product. Finally, the classifier is implemented as

$$F(x) = \arg \max_{k \in \{1, \ldots, M\}} \langle y^k, f(x) \rangle. \tag{3}$$

Note that the binary equations are the special cases of (2)-(3) for codewords $\{-1, 1\}$.

**Mutliclass Boosting:** MCBoost [34] is a multiclass boosting method that uses a set of unit vectors as codewords – forming a regular simplex in $\mathbb{R}^{M-1}$ –, and the exponential loss

$$L(z_i, f(x_i)) = \sum_{j=1, j \neq z_i}^M e^{-\frac{1}{2}[\langle y^{z_i}, f(x_i) \rangle - \langle y^j, f(x_i) \rangle]}. \tag{4}$$

For $M = 2$, this reduces to the loss $L(z_i, f(x_i)) = e^{-y_i z_i f(x_i)}$ of AdaBoost [9].

Given a set, $\mathcal{G}$, of weak learners $g(x) \in \mathcal{G} : \mathcal{X} \to \mathbb{R}^{M-1}$, MCBoost minimizes (1) by gradient descent in function space. In each iteration MCBoost computes the directional derivative of the risk for updating $f(x)$ along the direction of $g(x)$,

$$\delta R[f; g] = \left. \frac{\partial R[f + \epsilon g]}{\partial \epsilon} \right|_{\epsilon=0} = -\frac{1}{2n} \sum_{i=1}^n \langle g(x_i), w(x_i) \rangle, \tag{5}$$

where $w(x_i) = \sum_{j=1}^M (y^j - y^{z_i}) e^{-\frac{1}{2}\langle y^{z_i} - y^j, f(x_i) \rangle} \in \mathbb{R}^{M-1}$. The direction of steepest descent and the optimal step size toward that direction are then

$$g^* = \arg \min_{g \in \mathcal{G}} \delta R[f; g] \quad \alpha^* = \arg \min_{\alpha \in \mathbb{R}} R[f + \alpha g^*]. \tag{6}$$

The predictor is finally updated with $f := f + \alpha^* g^*$. This method is summarized in Algorithm 1. As previously mentioned, it reduces to AdaBoost [9] for $M = 2$, in which $\alpha^*$ has closed form.

**Mutliclass Kernel SVM (MC-KSVM) :** In the support vector machine (SVM) literature, the margin is defined as

$$\mathcal{M}(x_i, \mathbf{w}_{z_i}) = \langle \Phi(x_i), \mathbf{w}_{z_i} \rangle - \max_{l \neq z_i} \langle \Phi(x_i), \mathbf{w}_l \rangle, \tag{7}$$

where $\Phi(x)$ is a feature transformation, usually defined indirectly through a kernel $\mathbf{k}(x, x') = \langle \Phi(x), \Phi(x') \rangle$, and $\mathbf{w}_l$ ($l = 1 \ldots M$) are a set of discriminative projections. Several algorithms have been proposed for multiclass SVM learning [39, 44, 17, 5]. The classical formulation by Vapnik finds the projections that solve:

$$\begin{cases} \min_{\mathbf{w}_1 \ldots \mathbf{w}_M} & \sum_{l=1}^M \|\mathbf{w}_l\|_2^2 + C \sum_i \xi_i \\ s.t. & \langle \Phi(x_i), \mathbf{w}_{z_i} \rangle - \langle \Phi(x_i), \mathbf{w}_l \rangle \geq 1 - \xi_i, \forall (x_i, z_i) \in \mathcal{D}, l \neq z_i, \\ & \xi_i \geq 0 \quad \forall i. \end{cases} \tag{8}$$

---
**Algorithm 1 MCBoost**
---

**Input:** Number of classes $M$, number of iterations $N_b$, codewords $\{y^1, \ldots, y^M\} \in \mathbb{R}^{M-1}$, and dataset $\mathcal{D} = \{(x_i, z_i)\}_{i=1}^n$ where $z_i \in \{1 \ldots M\}$ is label of example $x_i$.

**Initialization:** Set $f = 0 \in \mathbb{R}^{M-1}$.

**for** $t = 1$ **to** $N_b$ **do**

    Find the best weak learner $g^*(x)$ and optimal step size $\alpha^*$ using (6).

    Update $f(x) := f(x) + \alpha^* g^*(x)$.

**end for**

**Output:** $F(x) = \arg\max_k \langle f(x), y^k \rangle$

---

Rewriting the constraints as

$$\xi_i \geq \max[0, 1 - (\langle \Phi(x_i), \mathbf{w}_{z_i} \rangle - \max_{l \neq z_i} \langle \Phi(x_i), \mathbf{w}_l \rangle)],$$

and using the fact that the objective function is monotonically increasing in $\xi_i$, this is identical to solving the problem

$$\min_{\mathbf{w}_1 \ldots \mathbf{w}_M} \quad \sum_i \lfloor \langle \Phi(x_i), \mathbf{w}_{z_i} \rangle - \max_{l \neq z_i} \langle \Phi(x_i), \mathbf{w}_l \rangle \rfloor_+ + \lambda \sum_{l=1}^M \|\mathbf{w}_l\|_2^2, \qquad (9)$$

where $\lfloor x \rfloor_+ = \max(0, 1 - x)$ is the hinge loss, and $\lambda = 1/C$. Hence, MC-KSVM minimizes the risk $R[f]$ subject to a regularization constraint on $\sum_l \|\mathbf{w}_l\|_2^2$. The predictor of the multiclass kernel SVM (MC-KSVM) is then defined as

$$F_{MC-KSVM}(x) = \arg\max_{l=1..M} \langle \Phi(x), \mathbf{w}_l^* \rangle. \qquad (10)$$

**Duality:** The discussion of the previous sections unveils an interesting duality between multiclass boosting and SVM. Since (7) and (10) are special cases of (2) and (3), respectively, the MC-SVM is a special case of the formulation of Section 2, with predictor $f(x) = \Phi(x)$ and codewords $y^l = \mathbf{w}_l$. This leads to the duality of Figure 1. Both boosting and SVM implement a classifier with a set of *linear* decision boundaries on a *prediction space* $\mathcal{F}$. This prediction space is the range space of the predictor $f(x)$. The linear decision boundaries are the planes whose normals are the codewords $y^l$. For both boosting and SVM, the decision boundaries implement a large margin classifier in $\mathcal{F}$. However, the learning procedure is different. For the SVM, examples are first mapped into $\mathcal{F}$ by a *pre-defined predictor*. This is the feature transformation $\Phi(x)$ that underlies the SVM kernel. The *codewords* (linear classifiers) are then *learned* so as to maximize the margin. On the other hand, for boosting, the *codewords are pre-defined* and the boosting algorithm *learns the predictor* $f(x)$ that maximizes the margin. The boosting / SVM duality is summarized in Table 1.

Table 1: Duality between MCBoost and MC-KSVM

|          | predictor      | codewords          |
|----------|----------------|--------------------|
| MCBoost  | learns $f(x)$  | fix $y_i$          |
| MC-KSVM  | fix $\Phi(x)$  | learns $\mathbf{w}_l$ |

## 3  Discriminant dimensionality reduction

In this section, we exploit the multiclass boosting / SVM duality to derive a new family of discriminant dimensionality reduction methods. Many learning problems require dimensionality reduction. This is usually done by mapping the space of features $\mathcal{X}$ to some lower dimensional space $\mathcal{Z}$, and then learning a classifier on $\mathcal{Z}$. However, the mapping from $\mathcal{X}$ to $\mathcal{Z}$ is usually quite difficult to learn. Unsupervised procedures, such as principal component analysis (PCA) or clustering, frequently eliminate discriminant dimensions of the data that are important for classification. On the other hand, supervised procedures tend to lead to complex optimization problems and can be quite difficult to implement. Using the proposed duality we argue that it is possible to use an embedding provided by boosting or SVM. In case of SVM this embedding is usually infinite dimensional which can make it impractical for some applications, e.g. hashing problem [20]. In case of boosting the embedding, $f(x)$, has a finite dimension $d$. In general, the complexity of learning a predictor $f(x)$ is inversely proportional to this dimension $d$, and lower dimensional codewords/predictors require more sophisticated predictor *learning*. For example, convolutional networks such as [22] use the

**Algorithm 2 Codeword boosting**

**Input:** Dataset $\mathcal{D} = \{(x_i, z_i)\}_{i=1}^n$ where $z_i \in \{1 \ldots M\}$ is label of example $x_i$, n. of classes $M$, a predictor $f(x) : \mathcal{X} \to \mathbb{R}^d$, n. of codeword learning iterations $N_c$ and a set of $d$ dimensional codewords $\mathcal{Y}$.
**for** $t = 1$ **to** $N_c$ **do**
    Compute $\frac{\partial R}{\partial \mathcal{Y}}$ and find the best step size, $\beta^*$ by (12).
    Update $\mathcal{Y} := \mathcal{Y} - \beta^* d\mathcal{Y}$.
    Normalize codewords in $\mathcal{Y}$ to satisfy constraint of (11).
**end for**
**Output:** Codeword set $\mathcal{Y}$

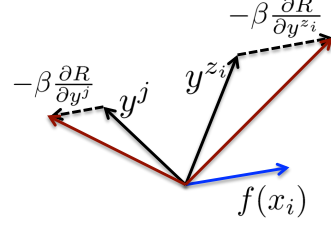

Figure 2: Codeword updates after a gradient descent step

canonical basis of $\mathbb{R}^M$ as codeword set, and a predictor composed of $M$ neural network outputs. This is a deep predictor, with multiple layers of feature transformation, using a combination of linear and non-linear operations. Similarly, as discussed in the previous section, MCBoost can be used to learn predictors of dimension $M$ or $M - 1$, by combining weak learners. A predictor learned by any of these methods can be interpreted as a low-dimensional embedding. Compared to the classic sequential approach of first learning an intermediate low dimensional space $\mathcal{Z}$ and then learning a predictor $f : \mathcal{Z} \to \mathcal{F} = \mathbb{R}^M$, these methods learn the classifier directly in a low-dimensional prediction space, i.e. $\mathcal{F} = \mathcal{Z}$. In the case of boosting, this leverages a classifier that *explicitly maximizes the classification margin* for the solution of the dimensionality reduction problem.

The main limitation of this approach is that current multiclass boosting methods [34, 27] rely on a *fixed* codeword dimension $d$, e.g. $d = M$ in [27] or $d = M - 1$ in [34]. In addition these codewords are pre-defined and are independent of the input data, e.g. vertices of a regular simplex in $\mathbb{R}^M$ or $\mathbb{R}^{M-1}$ [34]. In summary, the dimensionality of the predictor and codewords are tied to the number of classes. Next, we propose a method that extends current boosting algorithms 1) to use embeddings of arbitrary dimensions and 2) to learn the codewords (linear classifiers) based on the input data.

In principle, the formulation of section 2 is applicable to *any* codeword set and the challenge is to find the optimal codewords for a target dimension $d$. For this, we propose to leverage the duality between boosting and SVM. First, use boosting to learn the optimal predictor for a given set of codewords, and second use SVM to learn the optimal codewords for the given predictor. This procedure, has two limitations. First, although both are large margin methods, boosting and SVM use different loss functions (exponential *vs.* hinge). Hence, the procedure is not guaranteed to converge. Second, an algorithm based on multiple iterations of boosting and SVM learning is computationally intensive.

We avoid these problems by formulating the codeword learning problem in the boosting framework rather than an SVM formulation. For this, we note that, given a predictor $f(x)$, it is possible to learn a set of codewords $\mathcal{Y} = \{y^1 \ldots y^M\}$ that guarantees large margins, under the exponential loss, by solving

$$\begin{cases} \min_{y^1 \ldots y^M} & R[\mathcal{Y}, f] = \frac{1}{2n} \sum_{i=1}^n L(\mathcal{Y}, z_i, f(x_i)) \\ s.t. & \|y^k\| = 1 \quad \forall k \end{cases} \tag{11}$$

where $L(\mathcal{Y}, z_i, f(x_i)) = \sum_{j \neq z_i} e^{-\frac{1}{2}\langle y^{z_i} - y^j, f(x_i)\rangle}$. As is usual in boosting, we propose to solve this optimization by a gradient descent procedure. Each iteration of the proposed codeword boosting algorithm computes the risk derivatives with respect to all codewords and forms the matrix $\frac{\partial R}{\partial \mathcal{Y}} = \left[\frac{\partial R[\mathcal{Y}, f]}{\partial y^1} \ldots \frac{\partial R[\mathcal{Y}, f]}{\partial y^M}\right]$. The codewords are then updated according to $\mathcal{Y} = \mathcal{Y} - \beta^* \frac{\partial R}{\partial \mathcal{Y}}$ where

$$\beta^* = \arg\min_\beta R\left[\mathcal{Y} - \beta \frac{\partial R}{\partial \mathcal{Y}}, f\right], \tag{12}$$

is found by a line search. Finally, each codeword $y^l$ is normalized to satisfy the constraint of (11). This algorithm is summarized in Algorithm 2.

Given this, we are ready to introduce an algorithm that jointly optimizes the codeword set $\mathcal{Y}$ and predictor $f$. This is implemented using an alternate minimization procedure that iterates between the following two steps. First, given a codeword set $\mathcal{Y}$, determine the predictor $f^*(x)$ of minimum risk $R[\mathcal{Y}, f]$. This is implemented with MCBoost (Algorithm 1). Second, given the optimal predictor

---
**Algorithm 3 LADDER**

---

**Input:** number of classes $M$, dataset $\mathcal{D} = \{(x_i, z_i)\}_{i=1}^n$ where $z_i \in \{1 \ldots M\}$ is label of example $x_i$, number of predictor and codeword dimension $d$, number of boosting iteration $N_b$, number codeword learning iteration $N_c$ and number of interleaving rounds $N_r$.
**Initialization:** Set $f = 0 \in \mathbb{R}^d$ and initialize $\mathcal{Y}$.
**for** $t = 1$ **to** $N_r$ **do**
   Use $\mathcal{Y}$ and run $N_b$ iterations of MCBoost, Algorithm 1, to update $f(x)$.
   Use $f(x)$ and run $N_c$ iterations of gradient descent in Algorithm 2 to update $\mathcal{Y}$.
**end for**
**Output:** Predictor $f(x)$, codeword set $\mathcal{Y}$ and decision rule $F(x) = \arg\max_k \langle f(x), y^k \rangle$

---

$f^*(x)$, determine the codeword set $\mathcal{Y}^*$ of minimum risk $R[\mathcal{Y}^*, f^*]$. This is implemented with codeword boosting (Algorithm 2). Note that, unlike the combined SVM-Boosting solution, the two steps of this algorithm optimize the common risk of (11). Since this risk encourages predictors of large margin, the algorithm is denoted *Large mArgin Discriminant DimEnsionality Reduction* (LADDER). The procedure is summarized in Algorithm 3.

**Analysis**: First, note that the sub-problems solved by each step of LADDER, i.e. the minimization of $R[\mathcal{Y}, f]$ given $\mathcal{Y}$ or $f$, are convex. However, the overall optimization of (11) is not convex. Hence, the algorithm will converge to a local optimum, which depends on the initialization conditions. We propose an initialization procedure motivated by the following intuition. If two of the codewords are very close, e.g. $y^j \approx y^k$, then $\langle y^j, f(x) \rangle$ is very similar to $\langle y^k, f(x) \rangle$ and small variations of $x$ may change the classification results of (3) from $k$ to $j$ and vice-versa. This suggests that the codewords should be as distant from each other as possible. We thus propose to initialize the MCBoost codewords with the set of unit vectors of *maximum pair-wise distance*, e.g.

$$\max_{y^1 \ldots y^M} \min_{j \neq k} ||y^j - y^k||, \forall j \neq k \tag{13}$$

For $d = M$, these codewords can be the canonical basis of $\mathbb{R}^M$. We have implemented a barrier method from [18] to obtain maximum pair-wise distance codeword sets for any $d < M$.

Second, Algorithm 2 has interesting intuitions. We start by rewriting the risk derivatives as $\frac{\partial R[\mathcal{Y}, f]}{\partial y^j} = \frac{1}{2n} \sum_i (-1)^{\delta_{ij}} f(x_i) L_i s_{ij}^{(1-\delta_{ij})}$ where $L_i = L(\mathcal{Y}, z_i, f(x_i))$, $s_{ij} = \frac{e^{\frac{1}{2}\langle y^j, f(x_i) \rangle}}{\sum_{k \neq z_i} e^{\frac{1}{2}\langle y^k, f(x_i) \rangle}}$, and $\delta_{ij} = 1$ if $z_i = j$ and $\delta_{ij} = 0$ otherwise. It follows that the update of each codeword along the gradient ascent direction, $-\frac{\partial R[\mathcal{Y}, f]}{\partial y^j}$, is a weighted average of the predictions $f(x_i)$. Since $\delta_{ij}$ is an indicator of the examples $x_i$ in class $j$, the term $(-1)^{\delta_{ij}}$ reflects the assignment of examples to the classes. While each $x_i$ in class $j$ contributes to the update of $y^j$ with a multiple of the prediction $f(x_i)$, this contribution is $-f(x_i)$ for examples in classes other than $j$. Hence, each example $x_i$ in class $j$ *pulls* $y^j$ towards its current prediction $f(x_i)$, while *pulling* all other codewords in the opposite direction. This is illustrated in Figure 2. The result is an increase of the dot-product $\langle y^j, f(x_i) \rangle$, while the dot-products $\langle y^k, f(x_i) \rangle \, \forall k \neq j$ decrease. Besides encouraging correct classification, these dot product adjustments maximize the multiclass margin. This effect is *modulated* by the weight of the contribution of each point. This weight is the factor $L_i s_{ij}^{(1-\delta_{ij})}$, which has two components. The first, $L_i$, is the loss of the current predictor $f(x_i)$ for example $x_i$. This measures how much $x_i$ contributes to the current risk and is similar to the example weighting mechanism of AdaBoost. Training examples are weighted, so as to emphasize those poorly classified by the current predictor $f(x)$. The second, $s_{ij}^{(1-\delta_{ij})}$, only affects examples $x_i$ that do not belong to class $j$. For these, the weight is multiplied by $s_{ij}$. This computes a softmax-like operation among the codeword projections of $f(x_i)$ and is large when the projection along $y^j$ is one of the largest, and small otherwise. Hence, among examples $x_i$ from classes other than $j$ that have equivalent loss $L_i$, the learning algorithm weights more heavily those most likely to be mistakenly assigned to class $j$. In result, the emphasis on incorrectly classified examples is modulated by how much class pairs are confused by the current predictor. Examples from classes that are more confusable with class $j$ receive larger weight for the update of the latter.

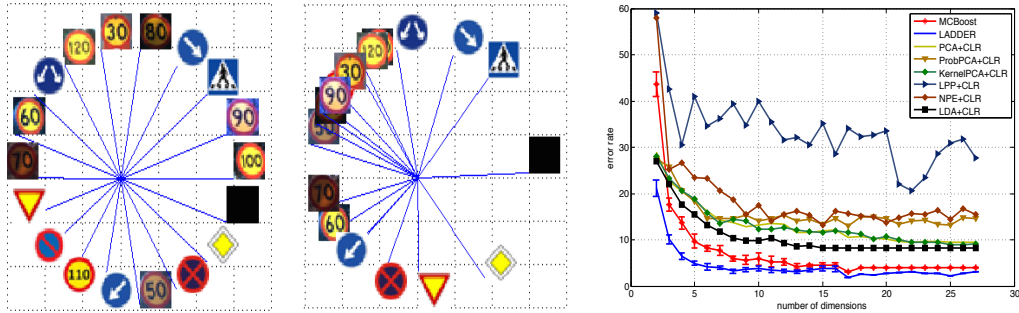

Figure 3: Left: Initial codewords for all traffic sign classes. Middle: codewords learned by LADDER. Right: Error rate evaluation with standard MCBoost classifier (CLR) with several dimensionality reduction techniques.

## 4 Experiments

We start with a traffic sign detection problem that allows some insight on the merits of learning codewords from data. This experiment was based on $\sim 2K$ instances from 17 different types of traffic signs in the first set of the Summer traffic sign dataset [25], which was split into training and test set. Examples of traffic signs are shown in the left of figure 3. We also collected about $1,000$ background images, to represent non-traffic sign images, leading to a total of 18 classes. The background class is shown as a black image in figure 3-left and middle. All images were resized to $40 \times 40$ pixels and the integral channel method of [8] was used to extract $810$ features per image.

The first experiment compared the performance of traditional multiclass boosting to LADDER. The former was implemented by running MCBoost (Algorithm 1) for $N_b = 200$ iterations, using the optimal solution of (13) as codeword set. LADDER was implemented with Algorithm 3, using $N_b = 2$, $N_c = 4$, and $N_r = 100$. In both cases, codewords were initialized with the solution of (13) and the initial assignment of codewords to classes was random. In each experiment, the learning algorithm was initialized with 5 different random assignments. Figure 3 compares the initial codewords (Left) to those learned by LADDER (Middle) for a 2-D embedding ($d = 2$). A video showing the evolution of the codewords is available in the supplementary materials. The organization of the learned codewords reflects the semantics of the various classes. Note, for example, how LADDER clusters the codewords associated with speed limit signs, which were initially scattered around the unit circle. On the other hand, all traffic sign codewords are pushed away from that of the background image class. Within the traffic sign class, round signs are positioned in one half-space and signs of other shapes on the other. Regarding discriminant power, a decision rule learned by MCBoost achieved $0.44 \pm 0.03$ error rate, while LADDER achieved $0.21 \pm 0.02$. In summary, codeword adaptation produces a significantly more discriminant prediction space.

This experiment was repeated for $d \in [2, 27]$, with the results of Figure 3-right. For small $d$, LADDER substantially improves on MCBoost (about half error rate for $d \leq 5$). LADDER was also compared to various classical dimensionality reduction techniques that do not operate on the prediction space. These included PCA, LDA, Probabilistic PCA [33], Kernel PCA [35], Locally Preserving Projections (LPP) [16], and Neighborhood Preserving Embedding (NPE) [15]. All implementations were provided by [1]. For each method, the data was mapped to a lower dimension $d$ and classified using MCBoost. LADDER outperformed all methods for all dimensions.

**Hashing and retrieval:** Image retrieval is a classical problem in Vision [3, 4]. Encoding high dimensional feature vectors into short binary codes to enable large scale retrieval has gained momentum in the last few years [6, 38, 23, 13, 24, 26]. LADDER enables the design of an effective discriminant hash code for retrieval systems. To obtain a $d$-bit hash, we learn a predictor $f(x) \in \mathbb{R}^d$. Each predictor coordinate is then thresholded and mapped to $\{0, 1\}$. Retrieval is finally based on the Hamming distance between these hash codes. We compare this hashing method to a number of popular techniques on CIFAR-10 [21], which contains 60K images of ten classes. Evaluation was based on the test settings of [26], using $1,000$ randomly selected images. Learning was based on a random set of $2,000$ images, sampled from the remaining 59K. All images are represented as $512$-dimensional GIST feature vectors [28]. The $1,000$ test images were used to query a database containing the remaining 59K images.

Table 2: Left: Mean average precision (mAP) for CIFAR-10. Right: Classification accuracy on MIT indoor scenes dataset.

| Method | hash length (bits) | | |
|---|---|---|---|
| | 8 | 10 | 12 |
| LSH | 0.147 | 0.150 | 0.150 |
| BRE | 0.156 | 0.156 | 0.158 |
| $ITQ_{unsup.}$ | 0.162 | 0.159 | 0.164 |
| $ITQ_{sup.}$ | 0.220 | 0.225 | 0.231 |
| MCBoost | 0.200 | 0.250 | 0.250 |
| KSH | **0.237** | 0.252 | 0.253 |
| LADDER | 0.224 | **0.270** | **0.266** |

| Method | Accuracy |
|---|---|
| RBoW [29] | 37.9% |
| SPM-SM [40] | 44.0% |
| HMP [2] | 47.6% |
| conv5+PCA+FV | 52.9% |
| conv5+MC-Boost+FV | 52.8% |
| conv5+LADDER+FV | **55.2%** |

Table 2-Left shows mean average precision (mAP) scores under different code lengths for LSH [6], BRE [23], ITQ [13], MCBoost [34], KSH [26] and LADDER. Several conclusions can be drawn. First, using a multiclass boosting technique with predefined equally spaced codewords of (13), MCBoost, we observe a competitive performance; on par with popular approaches such as ITQ, however slightly worst than KSH. Second, LADDER improves on MCBoost, with mAP gains that range from 6 to 12%. This is due to its ability of LADDER to adjust/learn codewords according to the training data. Finally, LADDER outperformed other popular methods for hash code lengths $\geq$ 10-bits. These gains are about 5 and 7% as compared to KSH, the second best method.

**Scene understanding:** In this experiment we show that LADDER can provide more efficient dimensionality reduction than regular methods such as PCA. For this we selected the scene understanding pipeline of [30, 14] that is consists of deep CNNs [22, 19], PCA, Fisher Vectors(FV) and SVM. PCA in this setting is necessary as the Fisher Vectors can become extremely high dimensional. We replaced the PCA component by embeddings of MCBoost and LADDER and compared their performance with PCA and other scene classification methods on the MIT Indoor dataset [32]. This is a dataset of 67 indoor scene categories where the standard train/test split contains 80 images for training and 20 images for testing per class. Table 2-Right summarizes performance of different methods. Again even with plain MCBoost predictor we observe a competitive performance; on par with PCA. The performance is then improved by LADDER by learning the embedding and codewords jointly.

## 5 Conclusions

In this work we present a duality between boosting and SVM. This duality is used to propose a novel discriminant dimensionality reduction method. We show that both boosting and K-SVM maximize the margin, using the combination of a non-linear predictor and linear classification. For K-SVM, the predictor (induced by the kernel) is fixed and the linear classifier is learned. For boosting, the linear classifier is fixed and the predictor is learned. It follows from this duality that 1) the predictor learned by boosting is a discriminant mapping, and 2) by iterating between boosting and SVM it should be possible to design better discriminant mappings. We propose the LADDER algorithm to efficiently implement the two steps and learn an embedding of arbitrary dimension. Experiments show that LADDER learns low-dimensional spaces that are more discriminant.

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
