[Reviews · NeurIPS 2016]

Reviewer 1

Summary

This manuscript derives a dimensionality reduction method. To get there, it shows a margin-based duality between SVMs and boosting.

Qualitative Assessment

This manuscript derives a dimensionality reduction method. To get there, it shows a margin-based duality between SVMs and boosting. The proposed discriminant based LADDER algorithm is analyzed correctly: the goal is to learn an as low dimensional as possible discriminant space. The experiments are limited, but the discussion is relevant on the sign domain. The authors should run a spell checker; there are a few minor typos.

Confidence in this Review

3-Expert (read the paper in detail, know the area, quite certain of my opinion)


Reviewer 2

Summary

The paper draws parallels (what the paper calls a "duality") between boosting and SVM's, and uses this connection to develop a new boosting-based technique for dimensionality reduction.

Qualitative Assessment

I liked the dimensionality-reduction approach suggested by this paper which seems sensible, fairly efficient, and effective in experiments. The "duality" with SVM's seems somewhat interesting. However, in the end, the approach that is taken is based only on boosting, rather than on an alternation between boosting and SVM's. So this connection is in fact only of limited benefit. The presentation is okay, but some of the math could be more precise. No real theory is given in this paper, such as generalization bounds, or proofs of convergence. Some theory might be possible. For instance, see Allwein et al. ("Reducing multiclass to binary..."). The alternating approach suggested in this paper seems reminiscent of (but is definitely different from) the approach given by Schapire ("Using output codes to boost multiclass learning problems"). A comparison might be of interest. Other comments: line 67: "tend to produce sub-optimal..." Give references of what evidence exists for this. eq (1): Of course, the approximate equality here only holds if uniform convergence bounds can be proved. And what is the expectation over? Also, the sum on the right should be multiplied by 1/n. line 94: Could omit the i subscripts in defining M(x_i, z_i), and elsewhere. In other words, just define margin M(x,z). That would simplify notation. lines 91 and following: This section could definitely be made more precise in terms of the math. line 95: Shouldn't the classifier map to {1,2}, not {-1,+1}? line 97: I think what is meant is that each y^z is in R^d. But what is written says that the set is in R^d. eq (7): Why the switch from z_i to c_i? Algorithm 2: Is {\cal Y} a set or a tuple? Shouldn't it be a tuple? And it is not an element of R^d, is it? This algorithm references eq (14), which does not seem to exist. eq (11): Should y^l be y^1? line 246: There does not appear to be such a figure 3b matching this description.

Confidence in this Review

3-Expert (read the paper in detail, know the area, quite certain of my opinion)


Reviewer 3

Summary

The paper notes that multiclass SVMs rely on a fixed mapping and learning a linear classifier. In contrast, multiclass boosting learns a mapping and applies a fixed classifier. The authors propose to learn both the mapping and the classifier. The paper is very well written, and fairly easy to follow. Experimental results appear sound. I am not an expert in either SVMs or Boosting, so I can't comment on novelty. If the work is novel, then I recommend acceptance.

Qualitative Assessment

As stated above, the authors note a duality between multiclass SVMs and Boosting and use this to propose a new algorithm that learns both a mapping and a classifier. In order to make this algorithm work, the authors replace the hinge loss of the SVM with an exponential loss (used by Booosting). So, to me, the proposed algorithm feels more like a Boosting algorithm, than a unification of Boosting and SVMs (but that's not a big deal). The algorithm solves a convex optimization problem in each iteration, so it does not solve a convex problem (I have no issues with this). The authors do not really discuss running time of the algorithm. It would be very good to provide numbers on the running time of the proposed algorithm and those of Boosting and SVMs. I think the work is interesting and the paper is well-written. I am, however, not an expert so I cannot judge importance or novelty. A few minor comments: * Sometimes you use \langle and \rangle for dot products, and sometimes you use < and >. Please be consistent. * You seem to use \argmin but not \argmax (rather you seem to use \arg\max) * When you talk of Vapnik's classic work, it would be good with a citation.

Confidence in this Review

1-Less confident (might not have understood significant parts)


Reviewer 4

Summary

The authors present an approach to dimensionality reduction for multi-class learning using a duality between SVMs and boosting. In particular, SVMs fix a feature transformation and learn the weights to separate the classes with decision boundary, while boosting fixes the codewords and learns the transformation. Based on this duality, a LADDER algorithm that iteratively updates both the codewords and the transformation in an efficient manner is developed. Several experiments were conducted to demonstrate the effectiveness of the proposed approach.

Qualitative Assessment

The paper is well written and easy to follow. My major concern of this paper is about the experiment part. For the hashing and retrieval part, the author did not compare their methods with some state-of-the-arts, e.g. SmartHash[1], FastHash[2], ShECC[3], OSH[4], and the one presented in [5], which they failed to beat. [1] Q. Yang, L.-K. Huang, W.-S. Zheng, and Y. Ling. Smart hashing update for fast response. In Proc. International Joint Conf. on Artificial Intelligence (IJCAI), 2013 [2] G. Lin, C. Shen, Q. Shi, A. van den Hengel, and D. Suter. Fast supervised hashing with decision trees for high-dimensional data. In Proc. IEEE Conf. on Computer Vision and Pattern Recognition (CVPR), 2014 [3] F. Cakir and S. Sclaroff. Supervised hashing with error correcting codes. In Proc. of the ACM International Conf. on Multimedia, 2014 [4] F. Cakir and S. Sclaroff. Online supervised hashing. In Proc. IEEE International Conf. on Image Processing (ICIP), 2015 [5] Cakir, Fatih, Sarah Adel Bargal, and Stan Sclaroff. "Online Supervised Hashing for Ever-Growing Datasets." arXiv preprint, 2015 Similar situation for the scene understanding experiment, where the state-of-the-art [6] has achieved over 80% accuracy on MIT indoor dataset and the proposed method only gave around 55%. The authors could at least try to integrate their method into a state-of-the-art approach to show if it will improve the performance. [6] Cimpoi, Mircea, Subhransu Maji, and Andrea Vedaldi. "Deep filter banks for texture recognition and segmentation." Proceedings of the IEEE Conference on Computer Vision and Pattern Recognition. 2015.

Confidence in this Review

2-Confident (read it all; understood it all reasonably well)


Reviewer 5

Summary

This paper deals with two major themes. First it describes a duality between multi-class SVMs and multiclass adaboost. Secondly it uses this duality to derive a novel dimensionality reduction method that preserves discriminability between different classes.

Qualitative Assessment

The authors modify the MCBoost criterion, in order to allow for multi-class boosting that is based on arbitrary number of dimensions (compared to a previous formulation that limits the number of dimensions to the number of classes). This lift of the limits in terms of dimensionality allows for a boosting-like framework that is comprised of controllable amount of boosting functions, and thus can be used as. + The connection between MC-Boost and MV-SVM is interesting, and the discussion is good. The discussion of the differences in terms of the predictor and the codewords (e.g. table 1) is also quite readable. Is the fact that both MC-SVM and MC-Boost try to maximise the margin well known? + The authors present improved results in terms of error rate, and in terms of mAP. More specifically, LADDER outperforms MCBoost by 2% in terms of classification accuracy. In addition, LADDER outperforms MCBoost for across all number of dimensions in terms of error rate (Figure 3). - More details could be given in terms of dimensionality reduction and CNN. This is a promising line of work, and has been shown to outperform all the other techniques even for cases where the size training data is not huge as the authors suggest. (e.g. DrLim Hadsell et al 2006). This holds for the literature review. - Results could be compared with the new results based on CNNs. The representations learned on CNNs can be quite powerful, since both the features and the discriminant projection is learned. In addition, altough CNN's are highly non-convex, in practice they give better results than convex methods. The only comparison with CNNs is given in terms of using the features extracted from conv5 (Table 2) in order to compare with the BoW features. However, the real power of deep learning comes from simultaneous learning of both the features and the discriminative projection functions. - Another convex method presented by Simonyan,Vedaldi and Zisserman (PAMI 2014) is not compared. In fact the authors only compare with lower accuracy methods such as PCA or LDA and their kernel equivalents. - In general the experimental evaluation is not convincing, and there are many more experiments needed to show the merits of the proposed method. Convergence characteristics and more sub-analyses of the proposed method are needed.

Confidence in this Review

2-Confident (read it all; understood it all reasonably well)


Reviewer 6

Summary

This paper suggests a method for learning SVMs for dimensionality reduction.

Qualitative Assessment

In abstraction, abbreviation LADDER should be explained (for example, Large Margin Discriminant Dimensionality Reduction (LADDER). In line 38, than -> that. Several math notations are wrong. In my review, the term 'predictor' is very confusing. Authors should explain the term 'predictor' (e.g. strong learner or weak learner). In equation 7, what is l and c_l?In line 178, I think the notation dY is wrong. In line 179, explanation for small letter 'm' is missing. What is f(x) in line 89? Is it real-valued function? Also, this paper needs to explain why the proposed algorithm correctly works in view of optimization.

Confidence in this Review

3-Expert (read the paper in detail, know the area, quite certain of my opinion)